# *Pentastiridius leporinus* (Linnaeus, 1761) as a Vector of Phloem-Restricted Pathogens on Potatoes: ‘*Candidatus* Arsenophonus Phytopathogenicus’ and ‘*Candidatus* Phytoplasma Solani’

**DOI:** 10.3390/insects15030189

**Published:** 2024-03-13

**Authors:** Eva Therhaag, Bernd Schneider, Kerstin Zikeli, Michael Maixner, Jürgen Gross

**Affiliations:** 1Institute for Plant Protection in Fruit Crops and Viticulture, Julius Kühn-Institut, Federal Research Centre for Cultivated Plants, 69221 Dossenheim, Germany; eva.therhaag@julius-kuehn.de (E.T.); bernd.schneider@julius-kuehn.de (B.S.); kerstin.zikeli@julius-kuehn.de (K.Z.); 2Institute for Plant Protection in Fruit Crops and Viticulture, Julius Kühn-Institut, Federal Research Centre for Cultivated Plants, 76833 Siebeldingen, Germany; michael.maixner@julius-kuehn.de

**Keywords:** *Pentastiridius leporinus*, potato, Stolbur, proteobacterium, transmission, gBlocks, choice trial, Cixiidae

## Abstract

**Simple Summary:**

In Germany, the planthopper *Pentastiridius leporinus* currently represents the most important vector for the spread of two pathogens, ‘*Candidatus* Arsenophonus phytopathogenicus’ and ‘*Candidatus* Phytoplasma solani’, which are associated with the syndrome “Basses Richesses” in sugar beets. In 2022, this planthopper was also found in potato fields in Germany along with symptoms of yellowing, wilting and rubbery tubers. In this study, greenhouse experiments with adult *P. leporinus* were conducted showing that it is able to transmit both pathogens to potatoes. Furthermore, it was confirmed that this vector can complete its entire lifecycle on both crops. To enable an accurate assessment of the phytoplasma titer in the vector and host plant, real-time PCR assays were performed using a synthetic DNA standard. Our study results highlight that *P. leporinus* may play an increasingly important role in agricultural cropping systems being a vector for two bacterial pathogens in two crops.

**Abstract:**

In Germany, the phloem-sucking planthopper *Pentastiridius leporinus* (Hemiptera: Cixiidae) currently represents the epidemiological driver for the spread of the syndrome “Basses Richesses” in sugar beets, which results in a reduced sugar content and an economic loss for the farmers. This disease is associated with the γ-proteobacterium ‘*Candidatus* Arsenophonus phytopathogenicus’ and the Stolbur phytoplasma ‘*Candidatus* Phytoplasma solani’. Recently, *P. leporinus* was found in potato fields in Germany and is associated with Stolbur-like symptoms in this crop. In this study, we confirmed that the vector completes its lifecycle on sugar beets as well as on potatoes when reared under controlled conditions. Transmission experiments with adults of this vector combined with molecular analyses showed, for the first time, that both pathogens are transmitted by this vector to potatoes. For an accurate assessment of the Ca. P. solani and Ca. A. phytopathogenicus titers in the vector and host plants, gBlocks derived from the *hsp*20- and 16S rRNA genes were employed, respectively. For Ca. P. solani, the limit of detection was determined in potato and sugar beet material. The results of this study will further the research on the epidemiology of the syndrome “Basses Richesses” and “Stolbur” diseases and the response of host plants and vector insects to both pathogens.

## 1. Introduction

The syndrome “Basses Richesses” (SBR) is a bacterial disease in sugar beets (*Beta vulgaris* ssp. *Vulgaris*), resulting in low sugar content of the beet root. In Germany, SBR was first reported in 2008 in the region of Heilbronn, Baden-Württemberg [1]. Since its first detection, it further spread across Germany and remains of high economic importance [1].

This disease is associated with the two vector-transmitted bacteria, ‘*Candidatus* Arsenophonus phytopathogenicus’*,* hereafter called CAp, and ‘*Candidatus* Phytoplasma solani’, hereafter called CPs [1,2,3,4,5]. CPs is a cell wall-lacking procaryotic microorganism of the provisional genus ‘*Candidatus* Phytoplasma’ within the class Mollicutes. This vector-born pathogen colonizes the plant phloem and is associated with numerous plant diseases [6]. CAp belongs to the γ subdivision of the class Proteobacteria [7]. Like CPs, the bacterium inhabits the plant phloem and is vector-transmitted by planthoppers of the family Cixiidae (Hemiptera) [8]. The only confirmed vector of SBR in sugar beets in Europe is the planthopper *Pentastiridius leporinus* [3,4,9].

In 2023, during the sugar beet harvest in Germany, a worsened situation was reported in SBR-affected regions, where about 60,000 ha of sugar beet fields had been infested by *P. leporinus* and infected by SBR, and another 10,000 ha in Southwest Germany and 4000 ha in Saxony-Anhalt with an increased incidence of rubbery taproots and the Stolbur phytoplasma in symptomatic beet roots [10,11,12]. The symptoms of the latter resembled that of rubbery taproot disease in sugar beets associated with CPs in Serbia, transmitted by the planthopper *Hyalesthes obsoletus* [13]. 

Recently, an increasing occurrence of potato plants (*Solanum tuberosum*) and tubers showing Stolbur-like symptoms such as yellowing, air tubers and rubbery tubers but also with symptoms of bacterial wilt, have been reported for about 8000 ha of potatoes grown in southwest Germany [12,14,15]. These symptoms are similar to those caused by CPs in potato in the Mediterranean and its northern range border in southwest Germany and north-east France (Alsace), where the Stolbur disease is also transmitted by the planthopper *H. obsoletus* [6,16]. 

The epidemiologically important vector for SBR in sugar beet in Germany and other countries, *P. leporinus* [2,3,4], has been reported in potato fields in Germany, firstly, in 2023 and is associated with yellowing and wilting symptoms [17]. A recent study by Behrmann et al. [15] showed that nymphs of *P. leporinus* fed and transmitted CAp to the potato, but the role of adults in transmitting the two SBR-associated pathogens to the potato has not been studied. In contrast to relatively immobile nymphs feeding below-ground, adults of this vector can fly and have a higher potential for migration, distributing the pathogens over longer distances. 

To study whether the potato may represent a potential feeding host for *P. leporinus*, especially in the vicinity of sugar beets and if so whether the vector can complete its entire lifecycle on potato like it does on sugar beets, choice and development experiments were performed in the present study. Furthermore, transmission experiments were carried out to elucidate whether adults of *P. leporinus* are capable of transmitting one or both pathogens to the potato. To quantify the CAp and CPs titer in plants and insects, TaqMan qPCR assays were performed employing a *hsp*20 gBlock [18] and a 16S rDNA-derived gBlock (this study). We also determined the limit of detection for CPs in potatoes and sugar beets.

## 2. Materials and Methods

### 2.1. Field Collection of P. leporinus

Planthoppers used in transmission trials and choice experiments were collected at the end of June 2023 with sweeping nets at two different locations in Southwestern Germany. The first site was a potato field near Bobenheim-Roxheim (49.5654220, 8.3914040), and the second site was a sugar beet field near Kirschgartshausen (49.5846780, 8.4544680). The insects were stored separately in insect cages for transport and identified by morphological traits [19]. Only *P. leporinus* specimen were used for the experiments, which started the same day.

### 2.2. Transmission Experiments

Adults of *P. leporinus* caught in a potato field were chosen for the transmission trial of five potato (*Solanum tuberosum*) varieties, namely ‘Lilly’, ‘Merle’, ‘Belana’, ‘Juventa’ and ‘Gala’. Certified planting tubers were used for the transmission trials. Individual tubers were planted on 9 June 2023 in pots and grown in an insect-proof greenhouse at 22 °C, and 14 days after germination, five plants per varieties were chosen for the transmission trials, as well as one control plant. All shoots but one were removed, and a foam plate was placed on top of the planting pot to cover the substrate. The shoots were 25–30 cm in height. An acrylic cylinder (height 35 cm, diameter 13 cm) covered with a mesh was placed 2 cm deep inside the substrate around the foam plate as a barrier. Ten planthoppers were transferred to each transmission cylinder, according to Jarausch et al. [20]. The planting pots with the cylinders were placed in a climate chamber with conditions slightly different to rearing protocols (22 °C, 55% relative humidity, and light/dark cycle of 18/6 h) [21,22]. The transmission experiments lasted for five consecutive days. Individuals that died during this period were removed and stored at −20 °C for further analysis, as well as the remaining individuals after the end of the transmission trial. Then, the cylinders were removed, and all plants were placed into an insect-proof greenhouse. The plants of the transmission trials were not sprayed or fumigated with any kind of pesticides because of not wanting to risk any effects on the plant–pathogen interaction. They were monitored for symptoms on a regular basis. Fifteen weeks post inoculation, all tubers per pot were harvested and cleaned with water. Three tubers per plant were tested individually for the presence of CPs and CAp by separate TaqMan assays.

### 2.3. Choice/Development Trial

*Pentastiridius leporinus* adults caught in a sugar beet field were subjected to a two-step choice trial to investigate host plant preferences and to determine the development speed on both host plants of potentially newly emerging offspring of the planthoppers. In the first step, each of the 35 adults were transferred to four net cages containing one sugar beet plant (variety ‘Vasco’, average height of 10 cm) and one potato plant (variety ‘Gerona’, average height of the shoots 20–30 cm). The 3 L pots were filled with potting soil and sand (0–2 mm diameter) in a ratio of 70:20 L. The substrate was covered with a 2 cm layer of expanded clay. The planthoppers were caged for two weeks. Insects that died during this time were removed and stored at −20 °C. After two weeks, all planthoppers still alive were removed and stored at −20 °C. In the second step, plants were rearranged. The four sugar beet plants were transferred to one cage while the potato plants were, due to their size, transferred to two cages. All plants were examined for the presence of eggs or nymphs. The cages were subsequently monitored for potentially emerging adults of the F1 generation three times per week for a period of six months in order to measure the development speed of *P. leporinus* on the different plant species. The trial took place in a climate chamber which was programmed, as in the transmission experiment for the first six weeks. After this time, the light period was set to 16/8 h light/dark mode, a temperature of 22 °C and 55% relative humidity for the remaining duration of the experiment. In the period of day 50 to day 90 of the trial, the temperature and the relative humidity differed dependent on the light/dark mode (24 °C/22 °C and 35%/55% for 21 days, and 24 °C/22 °C and 55%/70% for 14 days) due to a malfunction of the operating system.

### 2.4. DNA Isolation

DNA from plant material was extracted according to a modified Doyle protocol [23]. For the potato, about 50 mg of the navel (point of stolon attachment of the tuber), for sugar beet, 50 mg of the beet root, and for periwinkle (*Catharanthus roseus* (L.) G. Don), 50 mg of mid-ribs of the upper leaves were transferred into a microcentrifuge tube and homogenized with two stainless steel beads (hardened, non-rusting, 3.5 mm diameter, Kugel-Winnie, Bamberg, Germany) in a sample preparation system (Fast Prep^®^, MP Biomedicals, Irvine, CA, USA) for 60 s at speed level 4.0. Then, 500 µL of Doyle-buffer was added, and DNA extraction proceeded as described [23]. The final pellet was resuspended in 50 µL Tris-EDTA buffer (pH 8). Material from non-insect exposed potato plants (tubers or leaves) and sugar beets served as a negative control, whereas CPs-infected periwinkle served as a positive control for the Stolbur phytoplasma and CAp-infected sugar beets as a positive control for the γ-proteobacterium. 

DNA extraction from individual planthoppers was as follows: each planthopper was homogenized in a microcentrifuge tube containing 500 µL of Doyle buffer and 15 silica beads (SiLibeads^®^, type 2Y-S, 2.0–2.2 mm diameter, Sigmund Lindner GmbH, Warmensteinach, Germany) with the above-mentioned sample preparation system. All subsequent steps were according to the Görg et al. protocol [23].

The DNA yield for potato and sugar beet extracts was determined for three samples using a NanoPhotometer^®^NP80 (Implen GmbH, München, Germany) and by agarose gel electrophoresis using a dilution series of lambda DNA (New England Biolabs GmbH, Frankfurt am Main, Germany). For the prior measurement, the crude DNA extracts were treated with RNAse A (10 mg/mL, Fisher Scientific GmbH, Schwerte, Germany) and purified using a DNA purification kit (New England Biolabs GmbH, Frankfurt am Main, Germany).

### 2.5. Real-Time PCR Assays and Assessment of Sensitivity

Duplex TaqMan assays were employed for the detection of CAp and CPs in plant material. Both assays included a plant-specific primer and probe set to confirm a successful DNA extraction and an absence of inhibition [24]. Each reaction was comprised of 5 µL of qPCR PROBE Master Mix (primaQuant Probe Advanced, Steinbrenner Laborsysteme GmbH, Wiesenbach, Germany), 0.3 µL of each of the pathogen-specific primers (10 µM), the labelled probe (6-FAM/BHQ-1, 10 µM), 0.1 µL of each of the plant primers (10 µM) and the labelled plant probe (Cy5/Tamra, 10 µM), and 2.8 µL of nuclease-free HPLC-grade water, and one µL of the DNA sample was added. Sequences of the probes and primers are published [18,21,24].

For the detection of the pathogens in insect material, the reaction components were the same, except that the plant primers and probe were substituted with water. PCR conditions for the detection of CAp were 95 °C for 3 min, proceeded by 40 cycles of 95 °C for 15 s and 60 °C for 30 s. For CPs, the conditions differed slightly with 95 °C for 3 min, followed by 40 cycles of 95 °C for 10 s and 60 °C for 30 s. The threshold for positive detection was set at 35 cycles for both pathogens (cut-off value).

To determine the limit of detection for CPs in the potato and sugar beet, CPs-positive potato and sugar beet DNA extracts were serially diluted, and the decreasing DNA amount was replenished by DNA from healthy potatoes. A dilution series of the gBlock mixed with healthy DNA extracts was run in parallel. Real-time PCR assays were run in a Bio-Rad CFX96 Thermal Cycler (Bio-Rad Laboratories, Inc., Hercules, CA, USA).

### 2.6. Sequencing and Generation of a gBlock

A 16Sr DNA fragment of CPs from a phytoplasma-infected planthopper was amplified with primers SBRps_qRTforKL464/SBRps_qRTrevKL465 [21]. The amplimer was ligated in pGEM^®^-T (Promega Corporation, Madison, WI, USA) and transformed in *Escherichia coli* strain XL1 blue. The insert was sequenced with M13 universal and M13 reverse primers (Eurofins Genomics Europe Shared Services GmbH, Ebersberg, Germany). A 127 bp gBlock, identical to the insert, was synthesized by Integrated DNA Technologies (Leuven, Belgium). 

## 3. Results

### 3.1. Transmission Experiments

The transmission experiments revealed that the field-collected adult P. leporinus were able to transmit CPs as well as CAp to potatoes of various varieties. Of these, 48% of the transmission plants tested positive for CAp and 12% for CPs. No double-infection was detected. The results are shown in Table 1. Detailed information is provided in the Appendix A.

#### 3.1.1. Pathogens in Planthoppers

Each of the five plants per variety had been exposed to ten (in one case to eleven) planthoppers. At the end of the transmission period, a total of 222 planthoppers could be tested for infection by CAp and CPs. All planthoppers tested were CAp-positive in real-time PCR assays. The mean quantification cycle (C_q_) values differed, but 79.7% of the specimens were in the range of C_q_ 12–19, corresponding to a calculated number of 10^8^ to 10^6^ bacteria per insect. For CPs, only 6.3% of the planthoppers tested positive, and, consequently, only a fraction of the plants were actually exposed to CPs. The C_q_ values of eight individuals ranged between 16 and 19, corresponding to a calculated number of 5 × 10^6^ to 4 × 10^5^ organisms, and six individuals showed a C_q_ in the range of 25 to 32, which corresponds to about 0.5 × 10^4^ to 0.5 × 10^2^ organisms of the Stolbur phytoplasma per insect detected, respectively. 

#### 3.1.2. Pathogens in Plant Material

Before real-time PCR assays were performed, the DNA yield for potato and sugar beet extracts was determined. The DNA concentrations for both plants ranged between 20 and 35 ng/µL. All source material taken before the trial start was tested negative for both pathogens. After the transmission trial, the DNA of three tubers of each potato plant (*n* = 75), plus a negative control, were analyzed individually. Nineteen tubers tested CAp-positive (25.3%), and three tubers tested positive for CPs (4%). Except for the variety ‘Merle’, CAp was found in all varieties, whereas CPs infection was only identified in the variety ‘Merle’. However, one CPs-positive tuber was identified in a cage where all seven planthoppers tested negative for CPs, implying that at least one of the three missing insects was CPs-positive. Due to the low number of CPs-infected planthoppers and their heterogeneous distribution, no statement can be made on the potential susceptibility of the other varieties.

### 3.2. Choice Experiment

The inspection for deposited egg clutches took place two weeks after the start of the trial and revealed oviposition at the edge, bottom or top of the potted substrate. The number of eggs inside the clutches have not been counted, in order to prevent them from damage. Seven egg clutches were found in pots with potatoes (1; 3; 0; 3) and nine in pots with sugar beet plants (2; 2; 3; 2). Thus, no oviposition preference for one of the host plants could be detected.

### 3.3. Development Speed

Adults of the *P. leporinus* F1-generation emerged in the cages with potatoes, at the earliest 65 days (T_min_) and latest 77 days (T_max_) after the start of the experiment. Half of the adults (*n* = 17) emerged after 68 days (T_median_). In the cages with sugar beet plants, the first F1 adults emerged about two months later than in the potato. In total, eight adults emerged over time with T_min_, T_median_ and T_max_ being 125, 152 and 159 days, respectively (Figure 1).

### 3.4. Detection Limit of CPs Assay

For the quantification of CPs in plant and insect material, the sequence of the SBRps_qRTforKL464/SBRps_qRTrevKL465 amplimer was determined (Table 2) and used to synthesize a double-stranded DNA fragment (gBlock) of 127 base pairs. 

A ten-fold dilution series of the gBlock ranging from 1 × 10^8^ to 1 × 10^0^ revealed a reliable detection of 10 copies after 35 PCR cycles with the CPs TaqMan assay (Table 3). The C_q_ values between steps of 10 were 3.5 ± 0.57 apart. A copy number of one was not reliably identified with the setting used. Therefore, C_q_ readings after cycle 35 were considered as a negative result. 

The detection limit for CPs was determined for a symptomatic potato plant collected from an affected field (close to Heilbronn, Germany) and from a symptomatic sugar beet plant from the rearing of *P. leporinus* caught from a sugar beet field (near Heidelberg, Germany). They have been compared to the gBlock standard (Table 4). For the potato DNA extract, the C_q_ values (triplicates) ranged from 24.1 to 37.7 and for the sugar beet DNA extract from 22.1 to 36.8. The C_q_ values between the serial dilutions differed by an average of 3.4 (±0.15) for the potato, and 3.6 (±0.37) for the sugar beet. The C_q_ values of the serial dilutions of the gBlock mixed with potato or sugar beet DNA differed by an average of 3.1 (±1.6) and 3.6 (±1.3), respectively.

## 4. Discussion

Recently, potato growers in southwest Germany reported losses in the yield and tuber quality of plants, showing Stolbur-like symptoms [14,15]. This and the incidence of *P. leporinus* in potato fields in Germany in 2022 [17] prompted us to study the development of *P. leporinus* and its vectoring capacities for CPs and CAp in potato in more detail. Choice and transmission experiments were performed to elucidate its lifecycle on potatoes and to confirm its role as a potent vector.

Our study confirmed that potato plants represent a suitable host for *P. leporinus,* on which the insect can complete its entire lifecycle. We observed that the timespan from oviposition to the appearance of the first adults differed by almost two months between the development on the potato or sugar beet. However, the numbers of egg clutches and emerging adults were low, and thus, the results should be considered preliminary with regard to differences in the host’s plant-dependent development speed of the vector. The observation that the speed of the lifecycle depends on the host plant has been already observed for another cixiid planthopper, *H. obsoletus*. Here, a difference of the start of the flight activity was reported from populations developed on bindweed (*Convolvulus arvensis*), which approximately started three to four weeks prior to those that developed on nettle (*Urtica dioica*) [25]. However, the underlying factors such as the composition of the phloem sap have not yet been elucidated. Further research should be conducted to elucidate the phloem metabolites, which may be responsible for the differences in the development speed [26]. From the perspective of insect rearing, our results may be interesting, as they show that lifecycle experiments could be conducted in a shorter time. Pfitzer et al. reported a timespan from first the instar to adult of 193.6 ± 35.8 days for males and 193.5 ± 59.2 days for females on sugar beets, whereas Behrmann et al. reported a timespan of 170 days after hatching, also on sugar beets [21,22]. The implications of our finding for the expectable epidemiology of the transmitted diseases due to the possible increasing numbers of annual generations of the planthopper should also be studied in the future. 

In this study, we could further demonstrate that adults of *P. leporinus* are capable of transmitting CPs and CAp to potatoes. The fact that other cixiid species are able to vector CPs has been demonstrated before. In Southeastern Europe, *Reptalus quinquecostatus* transmits CPs to sugar beets [13], and *Reptalus panzeri* and *H. obsoletus* have been confirmed to transmit CPs to grapevines [27,28,29]. *Hyalesthes obsoletus* was further reported to be a vector for the Stolbur disease in lavender [30] but also in vegetables like tomato, eggplant, tobacco [31] and annual crops like maize [32], sugar beet [33] and potato [34,35,36,37,38,39], where it can cause serious effects in yield and tuber quality [40]. Whereas nymphs of *P. leporinus* have been reported to transmit CAp to potatoes [17], this is, to our knowledge, the first report that adults of this species transmit CPs as well as CAp to potatoes. Besides *P. leporinus*, no other planthopper species is known to vector CAp to plants, including sugar beets and potatoes.

It may not be appropriate to directly transfer the transmission results from lab experiments to field conditions, as the effects of landscape, tillage, insecticide use, or crop rotation may play a role for transmission of these pathogens to specific crops, but given the evidence that *P. leporinus* is well established in sugar beet cultivation and its major role in the epidemiology of SBR of sugar beets in Germany [2,3], data of this study indicate a high vectoring potential of *P. leporinus* for CPs and CAp to the potato. This may apply, in particular, to regions where this vector is already highly abundant and where both crops form part of the regional crop rotation and, thus, are simultaneously available as host plants. 

The transmission experiments showed another remarkable finding: none of the examined tubers of the variety ‘Merle’ tested positive for CAp, although all 42 insects were CAp-positive, showing high bacteria titers. Conversely, only three CPs-infected leafhoppers in three independent cages resulted in a successful transmission. Interestingly, the ‘Gala’ variety was not infected by CPs but by CAp, although insects were partly double-infected with comparable pathogen titers. These findings might indicate a differential susceptibility of potato varieties to CPs and CAp. 

So far, a correlation between the pathogen titer in the potato or sugar beet and the symptom severity is still missing. Therefore, a more comprehensive study should be conducted, including an evaluation of infection on yield and quality and on the susceptibility of varieties. These investigations have to be accompanied by real-time PCR monitoring, but caution is required on the pooling of sample material, since information is lacking on the spatial distribution of the two pathogens inside the tubers, stem or other plant tissue. 

With regards to real-time PCR diagnostics, different primers for CAp and CPs have been developed [18,41]. Behrmann et al. used real-time PCR assays for CAp and CPs based on 16S rDNA sequences with cloned quantification standards [21]. Since the production of a recombinant plasmid requires S1 security standards, we synthesized a gBlock derived from a 16S rDNA sequence for quantification, which can be used by plant protection services and other authorities working under lower security standards. With our settings, we were able to detect 10 copies of the DNA fragment, regardless of the gBlock being used as a template by itself or mixed with potato or sugar beet DNA. Lower copy numbers were not reliably identified. Both assays were suited to detect the pathogen with high sensitivity in picogram amounts of DNA. This quantification method, together with existing ones, may also facilitate further research on the response of the potato but also other plants to both pathogens. 

## 5. Conclusions

Our data present, for the first time, evidence that adults of *P. leporinus* transmit both CAp and CPs to the potato. We further show that *P. leporinus* can complete its whole lifecycle on this crop under controlled conditions. Our study also provides a new standard for real-time PCR, which allows for the quantification of the Stolbur pathogen in the tested plant or insect material. 

Further research is required on the varietal response of potatoes, such as tolerance, resistance, tuber quality and yield to pathogen titres and/or vector infestation. Also, effective and sustainable management options for this new vector–pathogen–plant complex must be investigated, as the affected potato-growing area in Germany and neighboring countries is likely to increase in the future, as we have already observed with the sugar beet.

## Figures and Tables

**Figure 1 insects-15-00189-f001:**
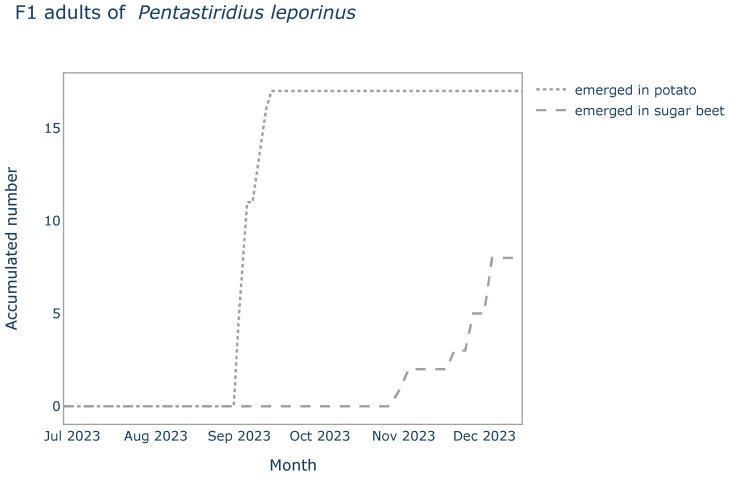
The lines represent data of emerged adults of the F1 generation of *P. leporinus* of the choice trial over time. Numbers have been accumulated per all pots (*n* = 4) of the same host plant (potato or sugar beet).

**Table 1 insects-15-00189-t001:** Summary of results of the transmission experiment from field-caught *P. leporinus* to five potato varieties. The percentage of CAp- and CPs-positive plants per variety was determined with real-time PCR analysis (five plants per variety, three tubers per plant).

Variety	Percentage of Plants Tested Positive for	
	CAp	CPs
Lilly	60%	0%
Merle	0%	60%
Belana	60%	0%
Juventa	40%	0%
Gala	80%	0%

**Table 2 insects-15-00189-t002:** Sequence of the 16SrDNA fragment of CPs amplified with primers SBRps_qRTforKL464/SBRps_qRTrevKL465.

Sequence (5′ to 3′)
ttgctaaagt ccccaactta atgatggcaa ttaacaacaa gggttgcgct cgttgcggga cttaacccaa catctcacga cacgagctga cgacaaccat gcaccacctg tgcttctgat aacctcc

**Table 3 insects-15-00189-t003:** Mean C_q_ values and their standard deviation (±SD) for the gBlock standards after qPCR.

Copy Number	C_q_ Values of the Target ^a^
1 × 10^0^	not detected (35.6 ^b^)
1 × 10^1^	35.3 (±0.51)
1 × 10^2^	32.8 (±0.81)
1 × 10^3^	28.6 (±0.37)
1 × 10^4^	25.0 (±0.15)
1 × 10^5^	21.8 (±0.51)
1 × 10^6^	18.3 (±0.36)
1 × 10^7^	14.7 (±1.00)
1 × 10^8^	10.8 (±1.51)

^a^ Mean of five technical replicates; ^b^ C_q_ value of one technical replicate; the other four technical replicates are not detected.

**Table 4 insects-15-00189-t004:** Determination of detection limits of C_q_ values of CPs in potato and sugar beet DNA extracts and of the gBlock standards.

Material	Dilution/Copy Number	C_q_ Values ^potato^	C_q_ Values ^sugar beet^
plant DNA	Positive sample	24.1 (±0.10)	22.1 (±0.17)
	1:10	27.3 (±0.10)	25.5 (±0.06)
	1:100	30.8 (±0.26)	29.0 (±0.12)
	1:1000	34.1 (±0.51)	32.5 (±0.17)
	1:10,000	not detected (37.7) ^a^	36.8 (±0.92)
	Negative sample	not detected (37.7) ^a^	not detected
gBlock with plant DNA ^b^	1 × 10^0^	37.5 ^c^ (±0.18)	not detected
	1 × 10^1^	36.6 (±1.02)	36.7 (±1.37)
	1 × 10^2^	34.7 (±0.49)	34.7 (±0.61)
	1 × 10^3^	29.2 (±0.04)	29.2 (±0.02)
	1 × 10^4^	26.3 (±0.07)	26.2 (±0.10)
	1 × 10^5^	22.4 (±0.54)	22.4 (±0.07)
	1 × 10^6^	18.9 (±0.11)	18.8 (±0.20)

Mean of three technical replicates of a CPs-positive potato or sugar beet DNA extract with DNA from non-infected potatoes or sugar beets, respectively. ^a^ C_q_ value of one technical replicate; the other two technical replicates were not detected. ^b^ gBlock standards diluted with potato DNA (2nd column) or sugar beet DNA (3rd column). ^c^ C_q_ value of two technical replicates; the other technical replicate was not detected.

## Data Availability

The original contributions presented in the study are included in the article, further inquiries can be directed to the corresponding author.

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
