# Peer review of "Pentastiridius leporinus (Linnaeus, 1761) as a Vector of Phloem-Restricted Pathogens on Potatoes: ‘Candidatus Arsenophonus Phytopathogenicus’ and ‘Candidatus Phytoplasma Solani’"

_insects, 2024, doi:10.3390/insects15030189_

Round 1

Reviewer 1 Report

Comments and Suggestions for Authors

Dear Authors,

It was a pleasure to read this manuscript, with very popular and ongoing subject of the research. Such research is very important at a time when agriculture plays a key role in providing enough food for a growing human population. This is an experimental work, involving many controlled factors, while at the same time providing very valuable results and answers that are more than applicative.

The manuscript is exquisitely written: each segment is concisely formulated, the research subject and results are clearly described and the discussion pointed to the importance of this study. Also, the number of the references is satisfactory, which are mostly recent publications.

Since genetic methods do not belong to my scope of research, I am not relevant to comment on that part, but the text is well written and based on the number of citations, I expect that the methods were used correctly and that the experiments were performed correctly.

A very few comments and suggestions are in the PDF document.

The only thing that I personally lack are some photos of plant samples and/or researched insect species during performing the experiment, purely for the sake of visualization and an even better understanding of the described experiments. In case you don't have photos for this research, please take this objection as an idea for some next manuscripts and experiments. 

Author Response

Thank you very much for taking the time to review this manuscript. Your feedback is very much appreciated. The three changes you suggested directly in the pdf of the manuscript are all done now. Other (minor) changes were added due to comments from other reviewers.

We are pleased to provide photos for a better visualization now in the supplementary materials.

Reviewer 2 Report

Comments and Suggestions for Authors

Title: Pentastiridius leporinus as a vector of phloem restricted pathogens on potato: ‘Candidatus Arsenophonus phytopathogenicus’ and ‘Candidatus Phytoplasma solani’

Review:
This study includes a few simple, but well excuted, experiments that showed for the first time evidence that adults of the planthopper P. leporinus transmit both CAp and CPs to potato plants, and that P. leporinus can complete its whole lifecycle on potato as well as sugarbeet. The ms. is acceptable (with minor revision) as a preliminary study as indicated in the discussion. The following suggestions/ clarifications, however, may improve the readability and accuracy of the current version.

Methods:
L. 108‐113. Did you fumigate or spray these plants with pestcides to kill any remaining or newly hatched planhoppers?!

Results:
L. 197. Replace "adults" with "field collected adults".
L. 198. Add percentage of infectd plants for each pathogen.
L. 198. Add "of various varieties" after "potato".

Table 1. This detailed and rather confusing table should be in supporting information. In the text, it should be replaced by a summary Table showing percentages of infected plants by both pathogens in various varieties.

Table 4. Top row, last 2 columns: Replace a and b with "potato" and "sugarbeet" respectively.
L.304. Delete "a" and add "and sugarbeet" after "potato".
L. 305-306. Delete "b to the end of this sentence.

End.

Author Response

This study includes a few simple, but well excuted, experiments that showed for the first time evidence that adults of the planthopper P. leporinus transmit both CAp and CPs to potato plants, and that P. leporinus can complete its whole lifecycle on potato as well as sugarbeet. The ms. is acceptable (with minor revision) as a preliminary study as indicated in the discussion. The following suggestions/ clarifications, however, may improve the readability and accuracy of the current version.

Response: Thank you very much for taking the time to review this manuscript. Your feedback is very much appreciated.

Methods:
L. 108‐113. Did you fumigate or spray these plants with pestcides to kill any remaining or newly hatched planhoppers?!

Response: The plants of the transmission trials were not sprayed or fumigated with any kind of pesticides not wanting to risk any effects on the plant-pathogen interaction. (Added: L. 117 -119)

Results:
L. 197. Replace "adults" with "field collected adults". Response: Thank you, it is done. (Now L.206)
L. 198. Add percentage of infectd plants for each pathogen. Response: Added, thank you. (Now L.207-208)
L. 198. Add "of various varieties" after "potato". Response: Ok, done. (Now L.207)

Table 1. This detailed and rather confusing table should be in supporting information. In the text, it should be replaced by a summary Table showing percentages of infected plants by both pathogens in various varieties. Response: Thank you for this suggestion. We changed it accordingly.

Table 4. Top row, last 2 columns: Replace a and b with "potato" and "sugarbeet" respectively.  Response: Thank you, it has been replaced. 
L.304. Delete "a" and add "and sugarbeet" after "potato". Response: We amended it to “or sugar beet” to clarify that we did not mix potato and sugar beet DNA together.
L. 305-306. Delete "b to the end of this sentence. Response: Done, thank you.
End.

Reviewer 3 Report

Comments and Suggestions for Authors

The manuscript by Therhaag et al. confirmed that planthopper (Pentastiridius leporinus) was able to transmit two pathogens, ‘Candidatus Arsenophonus phytopathogenicus’ and ‘Candidatus Phytoplasma solani’ to potato. Then they also detected the phytoplasma titer in the vector and host plant by real time-PCR assays. However, the manuscript has some issues that need to be resolved.  

 Comments:

 1The abstract seems lose some information, such as development of the planthoppers, and detection limit of CPs assay.

2I fail to see what the implication of “Choice experiment”, it seems not relate to this theme. In addition, in development trail, are CAp-positive or CPs-positive insects to measure the development speed? need explanation.

3The English grammar need to be checked, preferably by an native English speaker.

Comments on the Quality of English Language

The English grammar need to be checked, preferably by an native English speaker.

Author Response

Thank you very much for taking the time to review this manuscript. Your feedback is very much appreciated.

General

  • Are the methods adequately described? Can be improved. Response: Thank you. Photos of the transmission trials have been added in the supplementary material for a better visualization. 
  • Are the results clearly presented? Can be improved. Response: We agree and amended new summary Table 1 and 4 to improve the readability. Detailed information of Table 1 has been moved to supplementary materials (Table 1S).
  • Are the conclusions supported by the results? Can be improved. Response: Thank you. We hope that we could improve this with our explanation provided to your special comment (no. 2).

The manuscript by Therhaag et al. confirmed that planthopper (Pentastiridius leporinus) was able to transmit two pathogens, ‘Candidatus Arsenophonus phytopathogenicus’ and ‘Candidatus Phytoplasma solani’ to potato. Then they also detected the phytoplasma titer in the vector and host plant by real time-PCR assays. However, the manuscript has some issues that need to be resolved.  

Comments:

1、The abstract seems lose some information, such as development of the planthoppers, and detection limit of CPs assay.

Response: The number of words allowed in the abstract part is very limited so that we went deeper into these two technical matters only in the manuscript itself. Otherwise, we would have added this information to the abstract to comfort your comment.

2、I fail to see what the implication of “Choice experiment”, it seems not relate to this theme. In addition, in development trail, are CAp-positive or CPs-positive insects to measure the development speed? need explanation.

Response: Thank you for your questions. We conducted the choice trial since it was not known before, whether these planthoppers oviposit in potato (this was only known for sugar beet) when given a choice. Furthermore, we observed whether the planthoppers are able to fulfill their complete life cycle in both, potatoes and sugar beet, and if there are differences in developmental speed.

The pathogen status of the insects of the choice trial has not been analysed. Since both crops (sugar beet and potato) were available at the beginning of the choice trial, we cannot conclude which individual was laying eggs in which crop - regardless of its initial infection with one or both pathogens (CAp or CPs). Even if we had the insects tested, we could not trace them back to the plant (potato or sugar beet) they were laying eggs. This would have been different in a no-choice test.

3、The English grammar need to be checked, preferably by an native English speaker.

Sure, thank you.
